

# A new megaspilid wasp from Eocene Baltic amber (Hymenoptera: Ceraphronoidea), with notes on two non-ceraphronoid families: Radiophronidae and Stigmaphronidae

István Mikó[1], Thomas van de Kamp[2], Carolyn Trietsch[1], Jonah M. Ulmer[1], Marcus Zuber[2], Tilo Baumbach[2,3] and Andrew R. Deans[1]

[1] Frost Entomological Museum, Department of Entomology, Pennsylvania State University, University Park, PA, United States of America
[2] Laboratory for Applications of Synchrotron Radiation, Karlsruhe Institute of Technology (KIT), Karlsruhe, Germany
[3] Institute for Photon Science and Synchrotron Radiation, Karlsruhe Institute of Technology (KIT), Eggenstein-Leopoldshafen, Germany

Corresponding author
István Mikó, izm2@psu.edu

## ABSTRACT

Ceraphronoids are some of the most commonly collected hymenopterans, yet they remain rare in the fossil record. *Conostigmus talamasi* Mikó and Trietsch, sp. nov. from Baltic amber represents an intermediate form between the type genus, *Megaspilus*, and one of the most species-rich megaspilid genera, *Conostigmus*. We describe the new species using 3D data collected with synchrotron-based micro-CT equipment. This non-invasive technique allows for quick data collection in unusually high resolution, revealing morphological traits that are otherwise obscured by the amber. In describing this new species, we revise the diagnostic characters for Ceraphronoidea and discuss possible reasons why minute wasps with a pterostigma are often misidentified as ceraphronoids. Based on the lack of ceraphronoid characteristics, we remove *Dendrocerus dubitatus Brues, 1937*, Stigmaphronidae, and Radiophronidae from Ceraphronoidea and consider them as *incertae sedis*. We also provide some guidance for their future classification.

## INTRODUCTION

Ceraphronoidea is a hymenopteran lineage with an enigmatic phylogenetic position and poorly understood natural history. Their minute body size and parasitoid lifestyle, along with a few antennal and fore wing characters, suggest a close relationship with Proctotrupomorpha (*Ronquist et al., 1999*; *Engel & Grimaldi, 2009*). A myriad of other less-obvious morphological traits, however, including parts of the metasomal, genital and mesosomal skeletomuscular systems (*Vilhelmsen, Mikó & Krogmann, 2010*; *Mikó et al., 2013*; *Ernst, Mikó & Deans, 2013*), reveal many similarities to non-apocritan

Hymenoptera. Even recent molecular phylogenetic studies have failed to place the superfamily with confidence, although they support that the superfamily, indeed, is not closely related to Proctotrupomorpha (*Dowton et al., 1997*; *Heraty et al., 2011*; *Mao, Gibson & Dowton, 2015*; *Klopfstein et al., 2013*; *Peters et al., 2011*; *Peters et al., 2017*; *Branstetter et al., 2017*). Irrespective of their phylogenetic position, recent ceraphronoids comprise a morphologically well-characterized group that can be readily separated from other hymenopterans based on the following traits.

*1. Compact mesosoma.* In Ceraphronoidea, the pronotum, mesopectus, metapectus, first abdominal tergum, and the metanotum comprise a single, compact sclerite (Fig. 1A). This modification is only found in some wingless hymenopterans, as the presence of the conjunctivae that allows mobility between the above-mentioned mesosomal regions is less important in flightless hymenopterans (*Reid, 1941*; *Keller, Peeters & Beldade, 2014*). The functional consequences of mesosomal compactness in ceraphronoids have never been explored. Most ceraphronoids are able to fly; therefore, mesosomal compactness most likely evolved for a different reason than that of flightless hymenopterans with a similarly compact mesosoma.

*2. Orientation of the toruli and antennal bases.* The ventral position of the antennal insertion sites (toruli) is a well-known characteristic of Ceraphronoidea (*Masner, 1993*) and it is shared by some other apocritan taxa (e.g., Megalyroidea and Platygastroidea; *Sharkey et al. (2012)*). The orientations of the torular surface and the antennal base are, however, often overlooked traits of the superfamily. In Ceraphronoidea, the lateral torular margin is elevated relative to the median margin and therefore the antennal foramina, and the antennal scapes are oriented medially in a resting position (Figs. 1B, 1C).

*3. Articulation between pronotum and mesoscutum.* In Ceraphronoidea, the pronotum and the mesoscutum have a unique, ball-and-socket type articulation that corresponds to sharp anterolateral edges on both the mesoscutum and pronotum (Fig. 1A) (*Mikó & Deans, 2009*). The notauli arise from these articulation (Figs. 2A, 2B) in Ceraphronoidea. While distinct anterolateral edges on the mesoscutum are present in Megalyridae, they never correspond with ball-and-socket articulations.

*4. Wing venation.* All winged Ceraphronoidea have a stigmal vein that originates from the pterostigma or from the distal portion of the marginal vein posterior to the costal notch (*Masner, 1993*). Along with this, ceraphronoids have a single vein extending along the anterior margin of the fore wing. This vein is equipped with unique triangular elements (Figs. 1D, 1E), whose functions and origins remain unknown. A single vein on the anterior wing margin is present in numerous other hymenopterans (e.g., aphidiine braconids) but the triangular elements are seemingly specific to Ceraphronoidea (I Mikó, pers. obs., 2018). In many Chrysidoidea the two wing veins on the anterior wing margin (costal and subcostal veins) are adjacent and may superficially look ''fused'', but they are always separated by a faint line (*Olmi, 1994*; *Richards, 1939*).

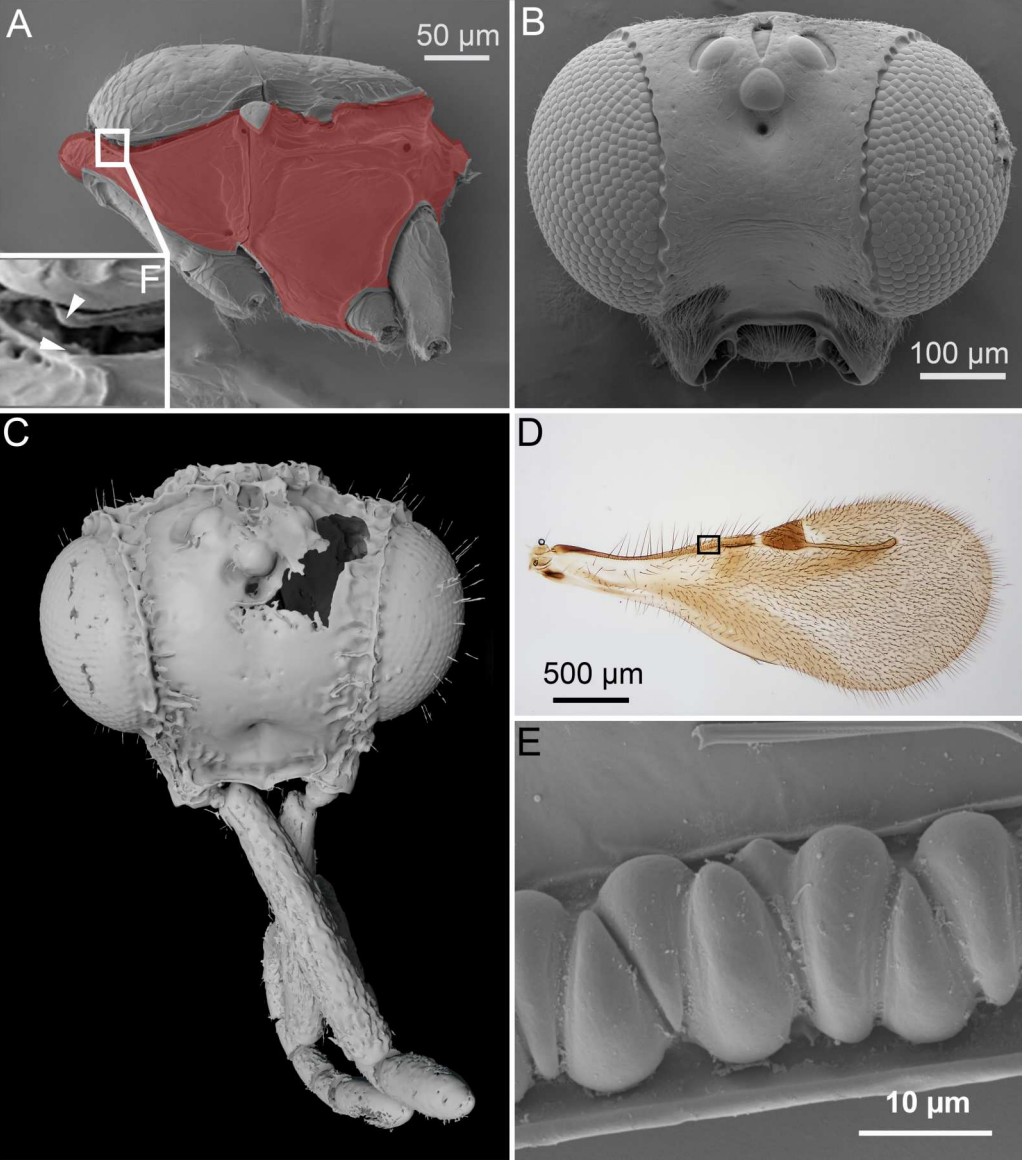

**Figure 1** **Unique ceraphronoid anatomical traits.** (A) SEM micrograph of the mesosoma of *Aphanogmus*. The pronotum, mesopectus, metapectus, metanotum and first abdominal tergum, shown in red, comprise a single sclerite resulting in a compact mesosoma that can be found only in wingless Hymenoptera (e.g., Formicidae workers and Mutillidae females). The mesonotum and the pronotum have a unique ball-and-socket type articulation (smaller box (F) marked with arrowheads), a trait shared by Ceraphronoidea and Megalyroidea. The function of this articulation is unknown. (B) SEM micrograph of the head of *Ceraphron* (subgenus *Eulagynodes*) sp. showing the medially-oriented antennal sockets (toruli). (C) Surface-rendered 3D reconstruction of the head of *Conostigmus talamasi* Mikó and Trietsch nov. sp, bearing medially-oriented toruli and scapes. (D) Brightfield image of the fore wing of *Masner lubomirus* Deans and Mikó 2009 showing the typical ceraphronoid wing venation. There is a single vein along the anterior wing margin equipped with triangular elements and an unbroken stigmal vein that arises from the posterior third of the pterostigma (the region of Figure E is indicated by a small box). (E) SEM micrograph showing the triangular elements on the fore wing of *Conostigmus* sp. The function of these unique elements is unknown.

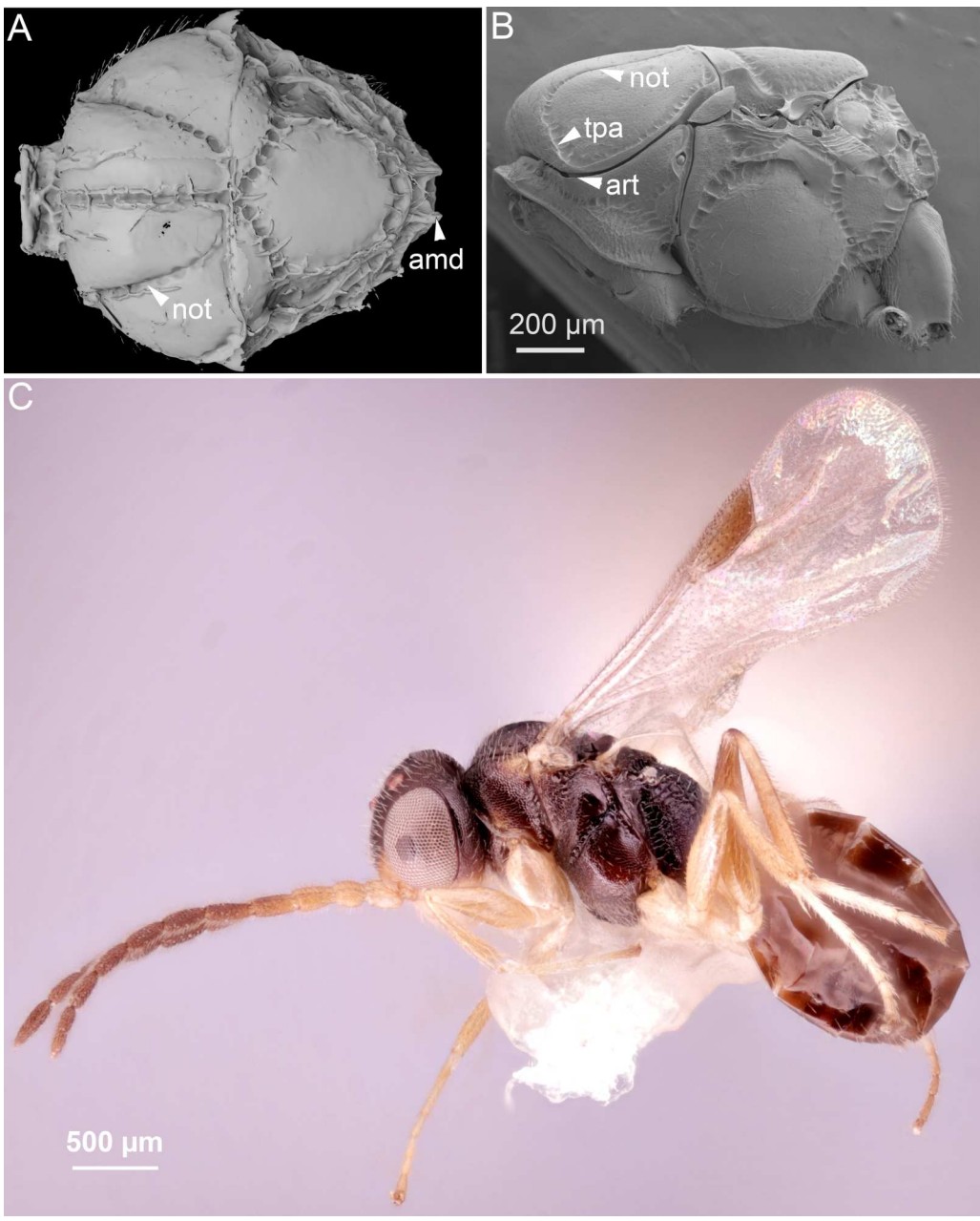

**Figure 2** **Comparison of ceraphronoid and chrysidoid morphology.** (A) Surface rendered 3D reconstruction of the mesosoma of *Conostigmus talamasi* Mikó and Trietsch nov. sp. showing the lyre-shaped notauli (not) and the bifurcated anteromedian projection of the metapectal-propodeal complex (holotype, DEI-GISHym31819). (B) SEM micrograph of the mesosoma of *Trichosteresis glabra* (Boheman, 1832). The anterolateral edge of the mesosoma corresponds to a ball-and-socket type articulation (art) between the mesonotum and the pronotum in Megalyroidea and Ceraphronoidea. The notaulus (not) is continuous with the anteromedian transverse sulcus (tpa) that arises from the pronoto-mesonotal articulation in Ceraphronoidea. (C) Brightfield image of the chrysidoid *Aphelopus* sp. These wasps, due to their minute body size, reduced wing venation, and large pterostigma, are often misidentified as Megaspilidae in collections (app, anteromedian projection).

The two type specimens of the present paper are embedded in an approximately 33–55 million-year-old (*Gillung & Winterton, 2018*; *Larsson, 1978*; *Penney, 2010*; *Sadowski et al., 2017*; *Wolfe et al., 2016*), somewhat cloudy piece of Baltic amber (Figs. 3C, 3D and 4A–4D). These specimens belong in Ceraphronoidea based on the the lyre-shaped notaulus and the orientation of the torulus and antenna (traits that are not obscured). As ceraphronoid experts, we examined the specimens with a light microscope at 230× magnification, and based on their general habitus (*Gestalt*) and the presence of a bifurcated anteromedian projection of the metanoto-propodeo-metapectal complex (Fig. 4A) we first classified them as *Megaspilus*, a genus that has never been recorded from any fossil deposit. However, after looking at the high resolution 3D data (see File S1), we were able to observe otherwise-obscured traits and determine that the new species actually belongs in *Conostigmus*, as part of a species group that shares some key features with *Megaspilus*.

While small body size, reduced wing venation, and the presence of a pterostigma are often used to determine specimens as Ceraphronoidea, these characters occur in almost all hymenopteran superfamilies. Despite the above-listed clear and obvious ceraphronoid synapomorphies, it is common to find smaller Ichneumonoidea and Aculeata, especially Bethylidae and Dryinidae, misidentified as Ceraphronoidea in collections. Perhaps the most commonly misidentified are small specimens of the dryinid genus *Aphelopus* (Fig. 2C). One factor that may contribute to these misidentifications is the use of low-power microscopes for sorting and identification of specimens. Even the most distinct external traits specific to ceraphronoids are obscure with lower magnification and inadequate lighting. In such poor conditions, only those who have trained their eyes by looking at hundreds of ceraphronoid specimens are able to identify these taxa correctly. In some cases, the examination of fossil specimens is similar to studying specimens of recent taxa with a low quality microscope. Morphological traits are usually obscured by artifacts or debris, making them difficult to properly observe (if they can be seen at all).

The discovery of these *Conostigmus* specimens encouraged us to review fossil ceraphronoids and revise the classification of two hymenopteran families that are exclusively represented by fossils and currently classified in Ceraphronoidea. Radiophronidae and Stigmaphronidae are minute, winged wasps that have a distinct pterostigma and, in some cases, two protibial spurs. Besides these traits, however, these two families do not share any other characteristics with Ceraphronoidea. Here, we remove them from the superfamily, leaving them *incertae sedis*, and provide some guidance for their future classification.

## MATERIALS AND METHODS

The two specimens for the present study were obtained from the private collection of Baltic amber inclusions of Christel and Hans Werner Hoffeins (Hamburg, Germany) who bought them from commercial source at Amberif in Gdansk. Specimens are embedded in Polyester resin (Voss-Chemie, Uetersen, Germany) (*Hoffeins, 2001*) and are deposited in the Deutsches Entomologisches Institut (Müncheberg, Germany) with the accession numbers: DEI-GISHym31819 (holotype), DEI-GISHym31820 (paratype).

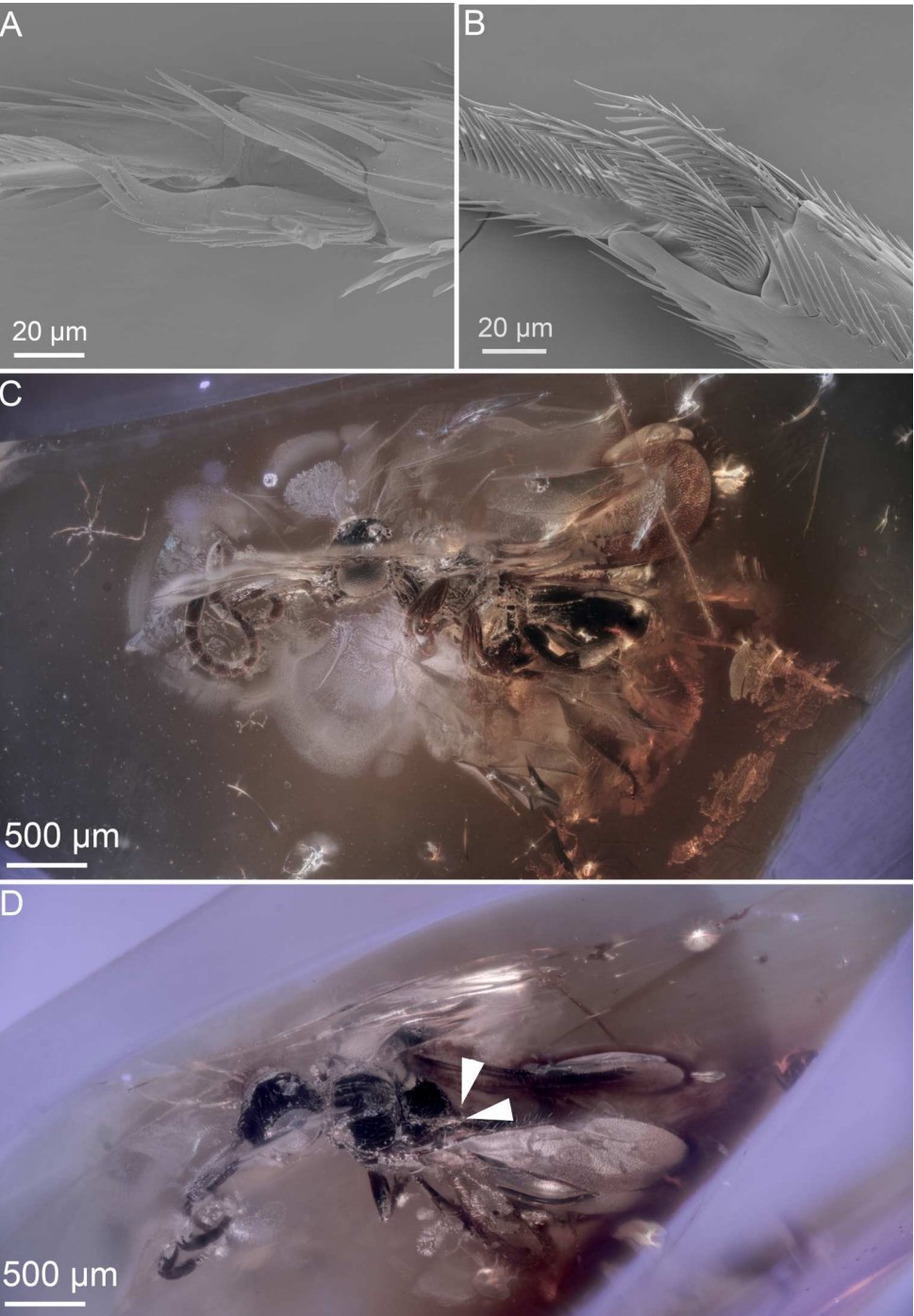

**Figure 3    Ceraphronoid morphology in recent and fossilized specimens.** (A) SEM micrograph showing the apical protibial spurs in Ceraphronidae. (B) SEM micrograph showing the apical protibial spurs in Megaspilidae. (C) Brightfield image showing the holotype of *Conostigmus talamasi* Mikó and Trietsch, lateral view (DEI-GISHym31819). (D) Brightfield image showing the holotype of *Conostigmus talamasi* Mikó and Trietsch, dorsolateral view, with arrows pointing to the bifurcate anteromedian projection of the metanoto-propodeo-metapectal complex (DEI-GISHym31819).

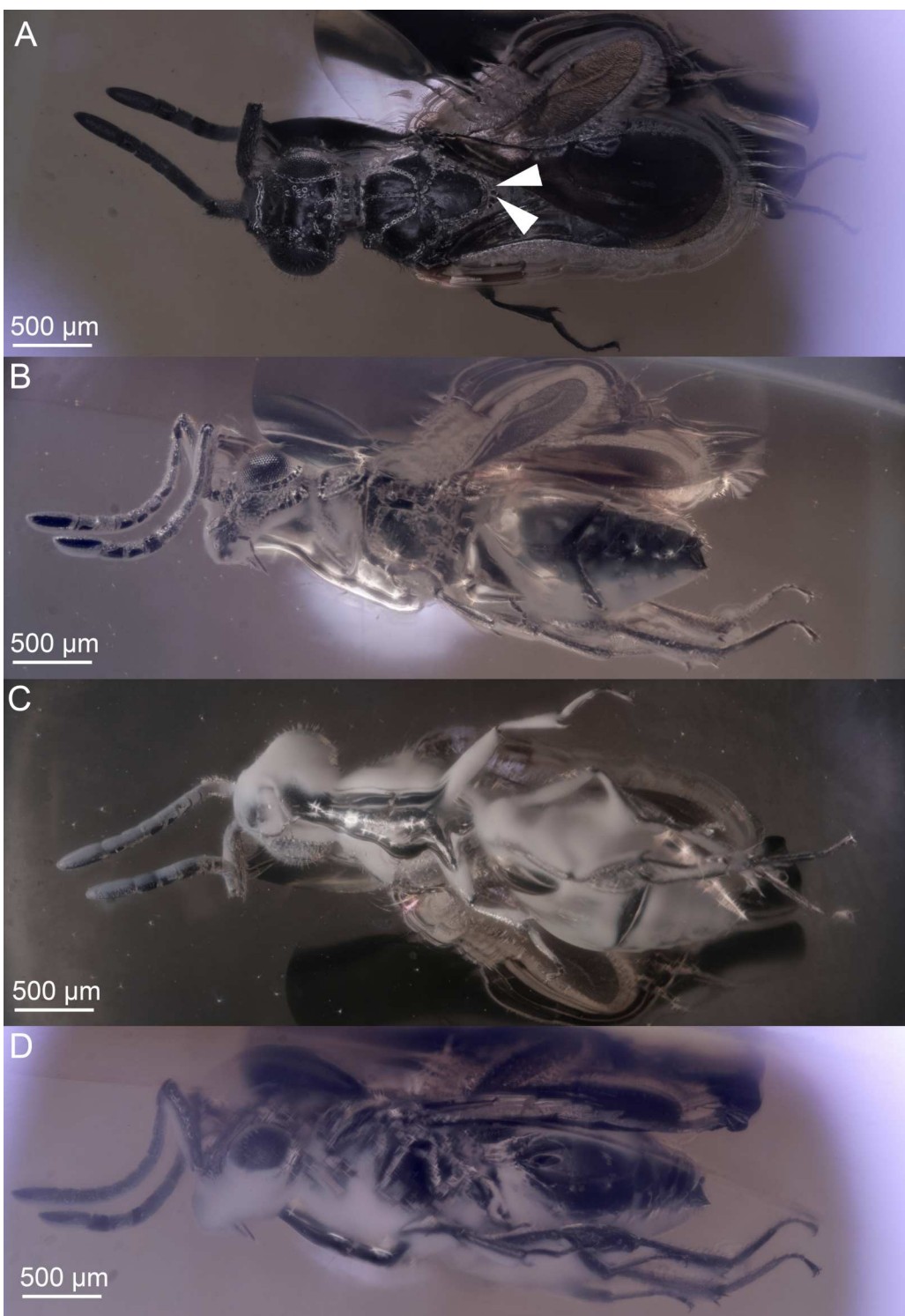

**Figure 4** **Brightfield images of the paratype (DEI-GISHym31820) of *Conostigmus talamasi* nov. sp. Mikó and Trietsch.** (A) Dorsal view, with an arrow pointing to the bifurcated anteromedian projection of the metanoto-propodeo-metapectal complex. (B) Left lateral view. (C) Ventral view. (D) Right lateral view.

Morphological traits were observed and described using volume and surface rendered 3D models (Figs. 1C, 2A, 5, Supplemental Information). Measurements of anatomical lines (Table S1) were performed using the 3D measurement tool in Amira (version 5.6, FEI) using the ASTOR virtual analysis infrastructure at Karlsruhe Institute of Technology (KIT) (*Mexner et al., 2017*).

Synchrotron X-ray tomography was performed at the UFO imaging station of the KIT light source. In order to achieve high image resolution despite the comparatively large size of the specimen, the latter was scanned in three steps. For each scan, 3,000 equiangularly-spaced radiographic projections were acquired in a range of 180°. The frame rate was set to 70 frames per second, resulting in a scan duration of about 43 s. A parallel polychromatic X-ray beam was spectrally filtered by 0.2 mm Al to obtain a peak at about 15 keV. The detector consisted of a thin, plan-parallel lutetium aluminum garnet single crystal scintillator doped with cerium (LuAG:Ce), optically coupled via a Nikon Nikkor 85/1.4 photo-lens to a pco.dimax camera with a pixel matrix of 2008 × 2008 pixels. The magnification of the optical system was adjusted to 10×, yielding an effective X-ray pixel size of 1.22 μm (*Dos Santos Rolo et al., 2014*). Tomographic reconstruction was performed with the GPU-accelerated filtered back projection algorithm implemented in the software framework UFO (*Vogelgesang et al., 2012*). The three tomographic volumes were registered and merged with Amira (version 5.6, FEI) using the ASTOR virtual analysis infrastructure at KIT (*Mexner et al., 2017*).

3D reconstruction followed the protocol described by *Ruthensteiner & Heß(2008)* and *Van de Kamp et al. (2014)*, using Amira for segmentation of every 20th slice of the tomographic volume. Automated interpolation between the labels was performed using the online image segmentation tool Biomedisa (https://biomedisa.de/) (*Lösel & Heuveline, 2016*). CINEMA 4D R18 (Maxon Computer GmbH) was employed for assembly of components, smoothing and polygon reduction. Subsequently, it was imported into Deep Exploration (version 6; Right Hemisphere), saved as Universal 3D file (U3D) and embedded into a PDF document with Adobe Acrobat 9 Pro Extended.

Brightfield images of fossil specimens were taken with an Olympus BX43 compound microscope equipped with an Olympus DP73 digital camera. Image stacking was performed with Zerene Stacker (Version 1.04 Build T201404082055; Zerene Systems LLC, Richland, WA, USA). Extended focus images were annotated and modified with Adobe Photoshop 6 (Adobe Systems, San Jose, CA, USA) using the Adjust/Filter/Unsharp mask and Image/Adjustments/Exposure (Gamma correction) tools.

Taxonomic treatment including natural language (NL) phenotype representations were compiled in mx (http://purl.org/NET/mx-database). Terminology of the phenotype statements used in descriptions, are mapped to the Hymenoptera Anatomy Ontology (HAO, available at http://purl.obolibrary.org/obo/hao.owl), Phenotypic Quality Ontology (PATO, available at http://purl.obolibrary.org/obo/pato.owl), Biospatial Ontology (BSPO, available at http://purl.obolibrary.org/obo/bspo.owl) and Common Anatomy Reference Ontology (CARO, available at http://obofoundry.org/). Wing venation terminology follows *Mikó et al. (2014)*.

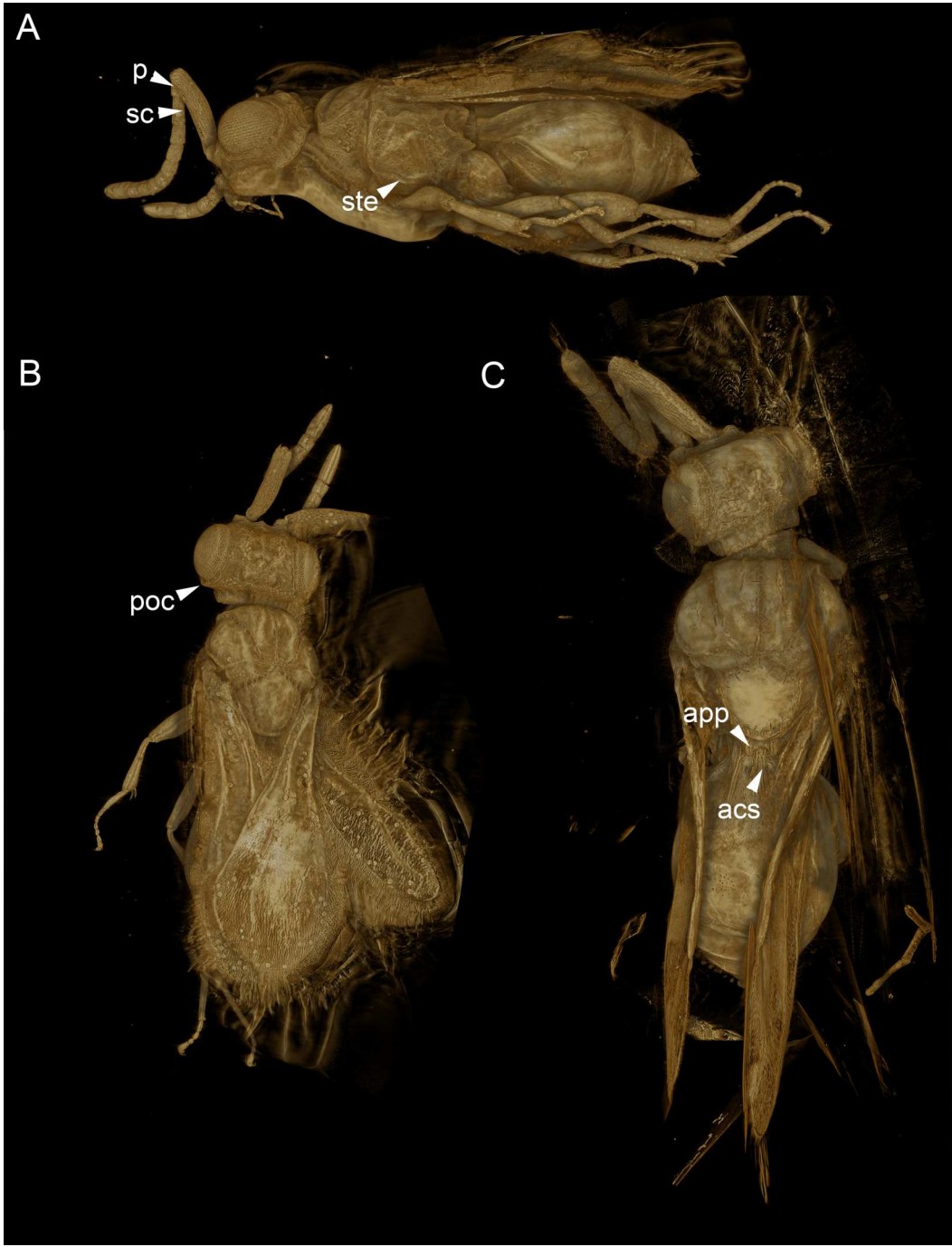

**Figure 5** **Volume rendered 3D micrographs showing *Conostigmus talamasi* sp. nov. Mikó and Trietsch.** (A) Paratype, lateral view. (B) Paratype, dorsal view. (C) Holotype, dorsal view. Abbreviations: acs, anteromedian carina of the syntergite; app, anteromedian projection of the metanoto-propodeo-metapecto-mesopectal complex; p, pedicel; poc, preoccipital carina; sc, scape; ste, sternaulus (DEI-GISHym31820).

Natural language phenotype representations are in "Entity attribute: value" format. Semantic statements written in OWL Manchester syntax (http://www.w3.org/TR/owl2-manchester-syntax/) were generated in Protégé 5.0.0-beta-16 (http://protege.stanford.edu/) following *Balhoff et al. (2013)*; *Mikó et al. (2014)*. The OWL (http://www.w3.org/TR/owl2-overview/; accessed February 4, 2014) representation of the full data set is stored as a single Resource Description Framework (RDF)-XML file (http://www.w3.org/TR/REC-rdf-syntax/; accessed 12 March 2017) in the Github repository (https://github.com/hymao/hymao-data).

The electronic version of this article in Portable Document Format (PDF) will represent a published work according to the International Commission on Zoological Nomenclature (ICZN), and hence the new names contained in the electronic version are effectively published under that Code from the electronic edition alone. This published work and the nomenclatural acts it contains have been registered in ZooBank, the online registration system for the ICZN. The ZooBank LSIDs (Life Science Identifiers) can be resolved and the associated information viewed through any standard web browser by appending the LSID to the prefix http://zoobank.org/. The LSID for this publication is: urn:lsid:zoobank.org:pub:0B233959-77FE-46F1-AB82-15C7F816D0BA. The online version of this work is archived and available from the following digital repositories: *PeerJ*, PubMed Central and CLOCKSS.

## RESULTS

### *Conostigmus*

The new species belongs in *Conostigmus* based on the presence of a distinct sternaulus (ste: Fig. 5A), a shorter posterior ocellar line (POL) than ocular-ocellar line (OOL) (Figs. 1C, 5B, 5C) and the presence of a preoccipital carina (Figs. 1C, 5A–5C). These traits are absent from *Dendrocerus*, *Trichosteresis*, and *Platyceraphron*; clavate female antenna and F1 length (proximodistal anatomical line) shorter than the combined length of F2 and F3 are traits that are present in *Conostigmus* and absent from *Megaspilus*.

### *Conostigmus talamasi* Mikó and Trietsch sp. nov.

urn:lsid:zoobank.org:act:B9777D1A-78D0-48B2-90F7-511DE4830EA9
Figs. 1C, 3B, 3C, 4, 5

*Diagnosis.* The new species differs from other *Conostigmus* species in the presence of a bifurcated anteromedian projection of the metanoto-propodeo-metapecto-mesopectal complex (app: Fig. 5C). The anteromedian carina of the syntergite (acs: Fig. 5C) is inserted into the concavity between the projections. The anteromedian projection of the metanoto-propodeo-metapecto-mesopectal complex in *C. talamasi* is distinct, but shorter than that of *Megaspilus* species.

*Description.* (Table S1)
Body length universal: 2007.29 µm, 2411.09 µm.
**Head:** Head width vs. head height: HW:HH = 1.183, 1.192. Head width vs. interorbital space (HW/IOS) Female: 1.861, 2.011. Dorsal carina of occipital depression count: present.

Dorsal carina of occipital depression medial continuity: discontinuous medially. Occipital carina sculpture: crenulate. Median flange of occipital carina count: absent. Submedial flange of occipital carina count: absent. Dorsal margin of occipital carina vs. dorsal margin of lateral ocellus in lateral view: occipital carina is ventral to lateral ocellus in lateral view. Preoccipital lunula count: present. Preoccipital ridge count: present. Preoccipital furrow count: present. Preoccipital furrow anterior extension: adjacent anteriorly to the posterior margin of the median ocellus. Preoccipital furrow anterior region vs posterior region sculpture: posterior region crenulate, anterior region smooth or finely reticulate. Preoccipital furrow anterior region width vs. posterior region width: as wide anteriorly as posteriorly. Preoccipital carina count: present. Preoccipital carina shape: interrupted dorsally and represented by irregular, not continuous carinae. Preoccipital carina and occipital carina structure: occipital carina complete, preoccipital carina fused laterally with preorbital carina. Female OOL: POL: LOL: 1.1:1.2:1,1.2:1.7:1. Postocellar carina count: absent. Preocellar pit count: present. Randomly sized areolae around setal pits on upper face count: absent. Antennal scrobe count: absent. Transverse striation on upper face count: present. Transverse scutes on upper face count: absent. Transverse frontal carina count: absent. Frontal ledge count: absent. Rugose region on upper face count: present. Anterior ocellar fovea shape: fovea not extended ventrally into facial sulcus. Facial pit count: facial pit present. White, thick setae on upper face count: absent. Ventromedian setiferous patch and ventrolateral setiferous patch count: absent. Supraclypeal depression count: absent. Intertorular carina count: present. Median process on intertorular carina count: present, extending from intertorular carina towards dorsal margin of clypeus. Median process of intertorular carina structure: process extends across intertorulal area to dorsal margin of clypeus. Intertorular ridge vs. epistomal ridge: fused medially. Intertorular area count: present. Median region of intertorular area shape: convex. Torulus position relative to anterior ocellus and distal margin of clypeus: torulus not reaching epistomal sulcus, closer to distal margin of clypeus than anterior ocellus. Torulo-clypeal carina count: absent. Subtorular carina count: absent. Subantennal groove count: absent. Posterolateral process of gena count: absent. Ocular impression and post ocular orbital carina count: present. Ocular impression sculpture: scalloped (foveae composing ocellar impression adjacent, sometimes not separated from each other). Mandibular tooth count: two. Mandibular lancea count: absent. Maxillary palpomeres count: five to six.

**Antennae:** Flagellar scrobe of the scape count: present. F1 length vs F2+F3 length: F1 shorter than F2+F3. F1 length / F2 length: 1.59, 1.55. F6 length / F7 length+F8 length: 0.59, 0.47. Scape length / F1 length+F2 length: 2.1, 2.25. Scape length / F2 length: 5.4, 5.74. Female scape length vs. pedicel length: 3.8,4.2. Female F1 length vs. pedicel length: F1 as long as pedicel (1.0–1.1). Female ninth flagellomere length: F9 longer than F7+8.

**Mesosoma:** Foveolate sculpture on body count: absent. Mesosoma shape: not compressed laterally, as wide as high or wider than high. Pronope count: present. Transverse pronotal sulcus (anterodorsal branch of pronotal y) count: present. Epomial carina count:absent. Posterodorsal branch of pronotal Y count: present. Ventrolateral invagination of the pronotum count: present. Annullar pronotum count: present. Ventromedian region of pronotum and anteroventral region of mesopectus continuity:

pronotum and mesopectus continuous ventromedially. Lateroventral invagination of the propleuron count: absent. Mesonotal fossa of the pronotum and pronotal condyle of the mesonotum count: present. Mesonotum anterolateral margin shape: square. Median mesoscutal sulcus count: present. Median mesoscutal sulcus posterior end location: adjacent to transscutal articulation. Scutoscutellar sulcus vs. transscutal articulation location: adjacent. Notaulus count: present. Notaulus posterior end location: adjacent to transscutal articulation. Posterior end of notaulus vs. posterior end of antero-admedian line location: notaulus extends more posteriorly than antero-admedian line. Transscutal articulation completeness: complete. Lateral carina on the mesoscutellum count: absent. Axillular carina count: present. Axillular carina shape: left and right carina continuous posteromedially forming a U-shape carina on the mesoscutellar axillar complex. Axillular setae count: absent. Posterolateral margin of mesoscutellum shape: blunt. Posteromedian process of the mesoscutellum count: absent. Anteromedian projection of the metanoto-propodeo-metapecto-mesopectal complex count: present. Anteromedian projection of the metathorax-propodeum complex shape: bilobed. Anteromedian projection of the metathorax-propodeum complex curvature lateral in view: straight. Sternaulus count: present. Sternaulus length: elongate, exceeding 3/4 of mesopleuron length at level of sternaulus. Longitudinal striae extending from crenulae of anterior mesopleural sulcus to mesopleural pit count: absent. Speculum ventral limit: not extending ventrally of pleural pit line. Mesometapleural sulcus count: present. Ventral invagination of mesometapleural sulcus presence: absent. Epicnemial carina count: complete. Epicnemial pit count: absent. Epicnemium posterior margin shape: anterior discrimenal pit present; epicnemial carina curved. Mesodiscrimen count: present. Anterior metapleural carina count: absent. Metapleural carina count: present. Metapleural carina vs. propodeal spiracle: metapleural carina extending ventrally of propodeal spiracle. Ventral projection of the metapleural carina count: present. Ventral invagination of the metapleural carina count: absent. Propodeal spiracle dilator muscle apodeme pit location: On metapleural carina. Lateral propodeal carina count: present. Lateral propodeal carina shape: inverted "U" (left and right lateral propodeal carina are adjacent to the antecostal sulcus of the first abdominal tergum submedially). Median propodeal carina count: absent. Posterior propodeal projection count: present. Posterior propodeal projection shape: simple. Propodeal and metacoxal verricules count: absent. Posterodorsal metapleural area shape: trapezoid. Posterior line of the posterodorsal metapectal area count: present. Transverse line of the metanotum-propodeum vs. antecostal sulcus of the first abdominal tergum: adjacent sublaterally. Carina limiting posteriorly antecosta count: present. Metapecto-propodeal conjunctiva count: present. Posterior margin of nucha in dorsal view shape: straight.

**Wings:** Stigmal vein of fore wing count: present. Pterostigma of fore wing count: present. Hind wing reduction: well developed.

**Legs:** Calcar shape: bifid. Mesotibial spur count: two. Mesobasicoxa width vs. metabasicoxa width: metabasicoxa distinctly wider than mesobasicoxa. Posterior mesosomal comb count: absent.

**Metasoma:** S1 length vs. shortest width: S1 wider than long. Transverse carina of petiole count: present. Transverse carina on petiole shape: straight. Basal, longitudinal carinae on

syntergum count: more than five. Transverse sulcus of first metasomal sternum count (S1 count): present. Waterston's evaporatorium count: absent.

## Locality of type specimens

Gulf of Gdańsk (Baltic amber)

## Etymology

The new species is named after Elijah Talamas (Florida State Collection of Arthropods), who drew our attention to these unique fossils.

## DISCUSSION

### *Megaspilus* vs. *Conostigmus*

Based on the latest phylogenetic analysis (*Mikó et al., 2013*), and preliminary phylogenomic data from an ongoing molecular study using ultra-conserved elements (UCEs) (B Blaimer, pers. comm., 2018) *Conostigmus* is polyphyletic and includes *Megaspilus*. Until Dessart's revisions (*1972*; *1981*) of Nearctic and Palaearctic species, *Megaspilus* was a broad taxonomic concept that essentially included all larger megaspiline species with an acute ocellar triangle (*Conostigmus*-type, in contrast to a *Dendrocerus*-type obtuse ocellar triangle), well-defined sternaulus, distinct posterior orbital carina and/or some rugulose sculpture on the frons. *Dessart (1972)* narrowed the generic concept of *Megaspilus* to include only those species that have an acute ocellar triangle, sternaulus, bifurcated anteromedian projection, and elongate female first flagellomere. The ocellar triangule shape and presence of the sternaulus are shared with numerous *Conostigmus* species, leaving the bifurcated anteromedian projection and the elongate female proximal flagellomeres as diagnostic features for *Megaspilus*.

The bifurcated anteromedian projection of *Megaspilus* is clearly derived from the elevated and medially-projected lateral propodeal carinae. This bifurcated condition can be found in certain *Ceraphron* and *Dendrocerus* (*Alekseev, 1978*) species, and so far has never been reported from any *Conostigmus* (*Mikó et al. 2016*; *Dessart 1997*; CT personal communictaion). The median portion of the lateral propodeal carina is elevated and forms a bifurcated projection in *C. talamasi* that is smaller than that of *Megaspilus* and most likely represents an intermediate state.

With the discovery of *C. talamasi*, the only diagnostic characters for separating *Megaspilus* from *Conostigmus* remain the elongate proximal female flagellomeres. The first female flagellomere is more than two times as long as the pedicel in *Megaspilus*, while in other megaspilids, including *Conostigmus*, it is less than 1.5 times as long as the pedicel. The length of the male and female flagellomeres correlate to each other and are important in species-level diagnosis in Megaspilinae (*Mikó et al., 2016*). The correlation of flagellomere length between different sexes might be related to their courtship behavior as males extensively antennate and repeatedly touch the female antenna during mating (*Liebscher, 1972*). Consequently, the first male flagellomere is much longer than the scape in *Megaspilus* in contrast with other male megaspilids where the first flagellomere is either shorter or slightly longer than the scape (*Mikó et al., 2016*; *Dessart, 1972*; *Dessart, 1981*; *Dessart, 1974*; *Dessart, 1995*; *Dessart, 1997*; *Dessart, 1999*; *Dessart, 2001*).

## Ceraphronoid and non-ceraphronoid fossils

To date, there are 18 fossil specimens that share key characteristics with recent Ceraphronoidea (see Fig. S1), including the compact mesosoma, lyre-shaped notaulus, and the orientation of the antennal bases. Based on their visible morphological characters, these fossils represent (mostly) megaspilid wasps from the Late Cretaceous Santonian to the Early Miocene. There is also a single ceraphronid specimen from the Early Miocene (Fig. S1).

Wing venation characters are perhaps the most distinctive features shared between these fossils and recent ceraphronoids. In these taxa, a single wing vein is present along the anterior fore wing margin and the stigmal vein is never angled (an angle is present in numerous other taxa in the stigmal vein, marking its intersection with 2RS or r-m veins *Mikó et al. (2014)*), never tangential to the pterostigma and arising from or anterior to the midpoint of the pterostigma (Fig. 1D, *Masner 1993*). A single fore wing vein can be found in numerous hymenopteran families, but unlike in these fossils, it is always well separated from the anterior margin.

The remaining fossil hymenopterans currently classified as ceraphronoids (Fig. S1) lack key ceraphronoid characteristics listed in the introduction. Although they do each possess a pterostigma, the remaining wing venation traits are inconsistent with the superfamily. These taxa might not even be closely related to Ceraphronoidea.

### *Dendrocerus dubitatus (Brues, 1937)*

*Dendrocerus dubitatus* (*Brues, 1937*) was the first described putative fossil ceraphronoid wasp. *Brues (1937)* explanation for his placement of this species is the following: ''This species is undoubtedly very similar to the large modern genus *Lygocerus.*'' Most of the body of the holotype specimen is obscured, however, and it is difficult to understand how Brues was able to prepare a rather detailed description as it was outlined by *McKellar & Engel (2011)*.

However, the wing venation on both fore wings are visible (Fig. 2A.), and they are cardinally different from that of recent ceraphronoids; the straight stigmal vein arises from the anterior portion of and is tangential with the pterostigma (Fig. 2A.). The antenna of *D. dubitatus* is composed of only 10 flagellomeres (Fig. 2B; *Brues 1937*), a character state that does not occur in any megaspilid taxa. While Ceraphronidae females often have 10 flagellomeres, the vast majority lack the pterostigma (*Trassedia* females have 11 flagellomeres, *Cancemi 1996*; *Mikó et al. 2013*; *Masner* is only known from male specimens, *Mikó & Deans 2009*). From these characters, it is clear that this species is not a ceraphronoid wasp. We consider it as *incertae sedis* and note that it has a wing venation often found in Chrysidoidea, as well as in some Stigmaphronidae.

### *Stigmaphronidae*

Another prospective ceraphronoid fossil, *Allocotidus* (*Muesebeck, 1963*), was described and classified with the following explanation (*Muesebeck, 1963* pg. 129): ''... the specimen ... is incomplete and otherwise in rather poor condition. Enough can be clearly made out, however, to place it in the proctotrupoid family Ceraphronidae.'' (Note that at the time of this description Megaspilidae and Ceraphronidae comprised a single family,

Ceraphronidae.) This taxon shares only one characteristic with ceraphronoids, the ''fused'' SC+R vein, although there is a faint line distinctly separating two wing veins at the anterior margin (a characteristic trait of numerous chrysidoids). *Kozlov (1975)* placed *Allocotidus* into his new family Stigmaphronidae, together with three new genera, *Stigmaphron* Kozlov, *Elasmomorpha* Kozlov and *Hippocoon* Kozlov, and provided a diagnosis largely based on traits shared with the Elasminae (flattened hind coxa, elongate tibial spurs, large mesoscutellar axillae complex, shortened metasoma). For reasons that remain unclear *Kozlov (1975)*, and subsequent authors of stigmaphronid taxa (*Engel & Grimaldi, 2009*; *Ortega-Blanco, Delclòs & Engel, 2011*; *McKellar & Engel, 2011*) considered these shared traits homoplasious, and classified Stigmaphronidae into Ceraphronoidea.

Stigmaphronidae show polymorphisms both in the number of protibial spurs and in wing venation characters. In some species, the two anterior fore wing veins are not adjacent to each other, the stigmal vein is either broken or arched or straight, and it arises anterior or in the middle of the pterostigma. The wing venation, however, never truly exhibits the characteristics of Ceraphronoidea. None of the stigmaphronid species shares any characteristics with recent ceraphronoids, except that some specimens have two protibial spurs. The presence of the two protibial spurs (Figs. 3A, 3B) has been long considered a plesiomorphic character state in Ceraphronoidea, despite evidence supporting the evolutionary plasticity of the number of tibial spurs in Apocrita (*Basibuyuk & Quicke, 1995*; *Kaartinen & Quicke, 2007*; *Engel & Grimaldi, 2009*). For example, even within Ceraphronoidea the mesotibial spurs are variable between Megaspilidae and Ceraphronidae.

The protibial spurs are particularly important in Hymenoptera systematics, as the anterior spur has evolved into an antenna cleaning device. This trait is an important synapomorphy for Hymenoptera (*Sharkey et al., 2012*; *Basibuyuk & Quicke, 1995*; *Vilhelmsen, Mikó & Krogmann, 2010*). The posterior spur has been reported as well developed, reduced, or absent in non-apocritans and is usually absent from apocritan taxa. There are known exceptions for two putatively unrelated braconid genera (*Rhamnura* and *Bathyaulax Basibuyuk & Quicke 1995*; *Kaartinen & Quicke 2007*) and recent Ceraphronoidea. This spur is difficult to differentiate from other apical, often unicellular protibial spines and trichoid sensilla (results of the evagination of the membrane of a single epidermal cell) in smaller specimens, even in recent taxa.

It is difficult for us to provide any guidance on how to reclassify stigmaphronids as, given the great polymorphism in tibial spurs, wing venation, and metasomal morphology (*Ortega-Blanco, Delclòs & Engel, 2011*), this taxon is likely polyphyletic. Based on the wing venation of Cretaceous chrysidoids, it is possible that stigmaphronids belong in Aculeata. A cenchrus-like area on the metanotum of one species (*Engel & Grimaldi, 2009*) suggests that at least this stigmaphronid might be closely related to some non-apocritan lineages. We consider Stigmaphronidae *incertae sedis*.

### Radiophronidae

As with Stigmaphronidae, the authors of Radiophronidae failed to provide a robust explanation for why they classified this family into Ceraphronoidea. The only character

state this taxon might share with Ceraphronoidea, besides the small body size, is the presence of two protibial spurs (*Ortega-Blanco, Rasnitsyn & Delclós, 2010*). These spurs, however, are difficult to observe in known specimens (Figs 1C1, 4C in *Ortega-Blanco, Rasnitsyn & Delclós, 2010*). Radiophronidae also lacks the most important ceraphronoid wing characteristics; they have two wing veins along the proximo-anterior margin of the fore wing instead of one, and the shape of the pterostigma is more elongate than in Ceraphronoidea. If the authors interpreted the fossil correctly, the pronotum of *Microstaphron* is visible in dorsal view and extends posteriorly, while the mesonotum is reduced. Similar modifications can be found in some *Ecnomothorax* (Megaspilinae), *Ecitonetes* (Ceraphronidae), and *Lagynodes* (Lagynodinae) species, but the enlarged pronotum and reduced mesonotum always corresponds to the reduction or absence of wings (*Brues, 1902*; *Dessart & Masner, 1965*; *Dessart, 1966*). Radiophronidae have well-developed wings. On the other hand, the pronotum is usually visible dorsally in Chrysidoidea, e.g., Bethylidae, which have similar wing venation to that of Radiophronidae (*Richards, 1939*).

Ceraphronoid male genitalia is unique among Apocrita in that they have an independent, moveable apical sclerite, the harpe *Mikó et al. (2013)*. The harpe is absent from the gonostyle-volsella complex of Radiophronidae (*Ortega-Blanco, Rasnitsyn & Delclós, 2010*). This condition–the absence of a harpe–can only be found in three distantly related recent ceraphronoid species: *Trichosteresis glabra*, *Aetholagynodes stupendus* and *Dendrocerus wollastoni* (*Mikó et al., 2013*).

Based on the wing venation, mesosomal, and male genitalia morphology, Radiophronidae most likely represents another unique lineage of Cretaceous chrysidoid wasps and should be considered *incertae sedis*.

## Is Ceraphronoidea the most structurally diverse hymenopteran superfamily?

Superfamilies, among the highest taxonomic ranks (family-level) that are regulated by the *ICZN (1999)*, serve as important taxa for communicating about Hymenoptera evolution (see *Sharkey et al., 2012*). They represent the highest functional and pragmatic taxa, defined, in part, by their natural history and a set of distinct morphological characteristics. It is critical, therefore, that they remain monophyletic and free from extraneous, unrelated taxa.

Recent and fossil Ceraphronidae and Megaspilidae, including *C. talamasi*, comprise a monophyletic group that can be clearly defined using numerous apomorphic traits (*Masner, 1993*). The inclusion of Stigmaphronidae, Radiophronidae and, until recently, the stephanoid Aptenoperissidae and the trigonaloid Maimetshidae (*Zhang et al., 2018*; *Perrichot et al., 2011*) within Ceraphronoidea, however, results in a polyphyletic morass that cannot be confidently diagnosed. The resulting taxon would have a variable number of protibial spurs and flagellomeres, the presence *and* absence of a wasp waist, tarsal plantulae, and cenchri, and either a compact or unabridged mesosoma, i.e., traits that are characteristic of other, well defined superfamilies. Ceraphronoidea has clearly been treated as a waste bin for minute fossil taxa with a pterostigma. Based on the characteristics of those fossils we remove these taxa from Ceraphronoidea.

## ACKNOWLEDGEMENTS

We give thanks to Elijah Talamas (Florida State Collection of Arthropods), who drew our attention to this unique fossil; to Christel and Hans Werner Hoffeins, who loaned us the specimens which are now permanently housed at the Deutsches Entomologisches Institut, Müncheberg, Germany (SDEI); Analytical tools used in this study were provided by the projects ASTOR and NOVA (Michael Heethoff, TU Darmstadt; Vincent Heuveline, Heidelberg University; Jürgen Becker, Karlsruhe Institute of Technology). We especially thank the following co-workers: Felix Beckmann, Jörg Hammel, Andreas Kopmann, Philipp Lösel, Wolfgang Mexner, Tomy dos Santos Rolo, Nicholas Tan Jerome, Matthias Vogelgesang, Tomáš Faragó, Sebastian Schmelzle. We acknowledge the KIT light source for provision of instruments at their beamlines and we would like to thank the Institute for Beam Physics and Technology (IBPT) for the operation of the storage ring, the Karlsruhe Research Accelerator (KARA).

### Funding

Research at KIT was partially funded by the German Federal Ministry of Education and Research (BMBF) by grants 05K2012 (UFO2), 05K2013 (ASTOR) and 05K2016 (NOVA). This material is based upon work supported by the US National Science Foundation, under Grant Numbers DBI-1356381 and DEB-1353252. There was no additional external funding received for this study. The funders had no role in study design, data collection and analysis, decision to publish, or preparation of the manuscript.

### Grant Disclosures

The following grant information was disclosed by the authors:
German Federal Ministry of Education and Research (BMBF): 05K2012 (UFO2), 05K2013 (ASTOR), 05K2016 (NOVA).
US National Science Foundation: DBI-1356381, DEB-1353252.

### Competing Interests

The authors declare there are no competing interests.

### Author Contributions

- István Mikó conceived and designed the experiments, performed the experiments, analyzed the data, prepared figures and/or tables, authored and reviewed drafts of the paper.
- Thomas van de Kamp conceived and designed the experiments, performed the experiments, analyzed the data, prepared figures, approved the final draft.
- Carolyn Trietsch analyzed the data, edited and approved the final draft.
- Jonah M. Ulmer and Marcus Zuber analyzed the data.
- Tilo Baumbach contributed reagents/materials/analysis tools.
- Andrew R. Deans contributed reagents/materials/analysis tools, approved the final draft.

## Data Availability

Github: https://github.com/hymao/hymao-data

Mikó, István (2018): 3D pdf of the surface rendered model of *Conostigmus* sp. nov. (Hymenoptera: Ceraphronoidea) from the Baltic amber. figshare. Figure. https://doi.org/10.6084/m9.figshare.5930455.v1

The specimens are permanently accessioned at the Deutsches Entomologisches Institut, Müncheberg, Germany (SDEI):

DEI-GISHym31819 HOLOTYPE *Conostigmus talamasi* Mikó & Trietsch

DEI-GISHym31820 PARATYPE *Conostigmus talamasi* Mikó & Trietsch.

## New Species Registration

The following information was supplied regarding the registration of a newly described species:

Publication LSID: urn:lsid:zoobank.org:pub:0B233959-77FE-46F1-AB82-15C7F816D0BA

Genus name: urn:lsid:zoobank.org:act:97EEB4F9-113A-41AD-93D7-175ABFEE64DF

Species name: urn:lsid:zoobank.org:act:B9777D1A-78D0-48B2-90F7-511DE4830EA9.

## Supplemental Information

Supplemental information for this article can be found online at http://dx.doi.org/10.7717/peerj.5174#supplemental-information.

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
