# Peer review of "A new megaspilid wasp from Eocene Baltic amber (Hymenoptera: Ceraphronoidea), with notes on two non-ceraphronoid families: Radiophronidae and Stigmaphronidae"

_PeerJ, doi:10.7717/peerj.5174_

## Round 0.1 · original submission · Minor Revisions

These comments and suggestions from the reviewers are very valuable. Please do make all of these requested changes as they will improve the scope and reach of the paper.

Reviewer 1 ·

Basic reporting

There are many references with incorrect spelling or formatting. I cannot list everything, but just a few examples:
- Bulletin-Institut royal (lines 423, 428) vs. Bulletin de l’Institut Royal (line 420)
- all references from Dessart lack the French accents. Ex. for 1966: should be ‘Contribution à l’étude des Hyménoptères …’
- Genus/species names not italicized in many cases…
- the use of upper case and lower case is a real mess.

Also, some figures are not appropriately labeled (see details in the general comments below).

Experimental design

no comment

Validity of the findings

no comment

Additional comments

I appreciate the effort of the authors for the detailed description and illustration of the new fossil species, coupled with the revision of taxa from a group of wasps with a scant fossil record. Particularly useful for future studies is the brief review of recent ceraphronoid’s diagnostic traits provided in introduction. Also, the description of the new species and suggested placement of D. dubitatus, Radiophronidae and Stigmaphronidae as incertae sedis are convincing. In clear, the study is sound and is certainly worth the publication in PeerJ.

The MS suffers many factual errors or inaccuracies, however, that make its reading sometimes unpleasant or difficult to follow. Below are my comments on things that need revision or complement before the MS can be published:

- Line 1, title: when dealing with a fossil taxon, it is important to provide the age in the title. (i.e., "A new megaspilid wasp from Eocene Baltic amber »;
- Line 27 and throughout: Dendrocerus and Lygocerus are alternatively used for the species dubitatus. Even in line 307, D. dubitatus comes out among the Lygocerus paragraph. Please be consistent and use the valid combination only!
- Line 41: "pronotum, mesopectus, metapectal-propodeal complex and metanotum" used here, but metapectus used in Fig. 1. Be consistent.
- Line 69: Baltic amber is NOT 55 myo! Where does this age come from? Correct age is Middle to Late Eocene (Lutetian-Priabonian, 34-48 Ma according to the International stratigraphic chart, latest version 2017/2: http://www.stratigraphy.org/index.php/ics-chart-timescale). See Sadowski et al. 2017 (in: Am. J. Botany 104: 694-518; or in Stapfia 106: 1-76) and references therein for further details and discussion on the age.
- Line 70: the studied amber piece, and pieces of Baltic amber in general, cannot be called ‘opaque amber’ in any way. Baltic amber is distinctly translucent compared to amber from some other deposits. The opaqueness as seen by the authors results from examination at high resolution with a light microscope, without both incident and transmitted lights.
- line 74: the bifid anteromedian projection is absolutely invisible on Fig. 4A. Better refer to the Supplementary 3D figure only.
- Lines 79-90: I am OK with this assertion, but same is true for entomologists like the authors working punctually on a fossil (whether an amber inclusion or rock imprint): they have clearly no knowledge on fossil deposits, chrono- or biostratigraphy, geological ages, and examination methods for such material. Here and in couple previous papers, the authors claim that a correct study of fossil wasps can only be made by neontologist experts like them, but they should seriously consider adjoining the collaboration of a paleoentomologist to avoid basic mistakes in the interpretation of fossils and their geological record.
- Caption of fig. 1:
1/ a similarly compact mesosoma can also be found in the extinct, Cretaceus family Maimetshidae (previously suggested as a possible ceraphronoid family, by the way), though these wasps are always winged (but no idea if the ball-and-socket articulation is present in these wasps…). Discussion on this family should be added.
2/ In (D), should read ceraphronoid instead of cerahronoid
3/ "an unbroken pterostigma that arises from the posterior third of of pterostigma" ? Makes no sense to me…
- Fig. 2A: too dark, please improve the contrast (i.e., no dark blue on a black background); also the bifurcated projection should be indicated by an arrow.
- Lines 101-102: accession numbers do not correspond to those in Table S1( i.e, #1013 and 1805). Be consistent.
- Lines 106-108: meaning of KIT is explained in its second use only (line 108); should be explained in first use. Same for POL and OOL in line 161
- Line 116: when speaking of 3D tomography, I think the correct word is voxel, not ‘pixel’.
- Lines 107-126: not what I would call a method of ‘quick data collection’ (cf. abstract)!
- Line 161: fig 1B is not Conostigmus. Do you mean fig 1C?
- Line 163: note that F1 length / F2+F3 length are not given in description nor in Table S1. They should appear somewhere.
- Fig 3D: the bifid projection is not visible at this scale and under this view. Thus, arrows are entirely useless.
- Figs 4B-D: seems to be the paratype rather than the holotype; also for 4D: not a right lateral view – seems the same as 4B to me.
- Fig. 5B: meaning of the arrow on the head? Not explained in caption
- Fig5 caption: except for the scale and pedicel, the structures given in abbreviations are nearly invisible at this scale. Better refer to Supplemental 3D figure.
- The call to figures is incorrect in several cases (e.g, line 53: Figs 1A, B should be Figs 1B, C; line 167: Figures 1C, 3B, C, 4, 5 should be 1C, 2A, 3C, 3D, 4, 5, etc…). Check all calls to figures carefully.
- Line 173: are these two metrics the holotype and paratype lengths? (cf. Tab S1, specimen numbers unclear).
- Line 269 and throughout: references are often incorrectly formatted, e.g.: Conostigmus (Mikó et al. (2016); Dessart (1997) … should read (Mikó et al., 2016; Dessart, 1997; CT personal observation). Check throughout the entire MS.
- Line 281: years of references should read consecutively, e.g.: Dessart 1972 to 2001.
- Line 310: should read pterostigma, not pterostigmata
- Line 386: delete one ‘confidently’
- Line 400: add caption of supplementary table 1.
- Ref. Zhang et al. 2017 is now published; Please update with volume and page numbers.
- Suppl. Fig 1:
1/ Abbreviations (S, A, R, M, C) are not explained.
2/ D. dubitatus is not listed!
3/ Correct name is Burdigalian, not Burdigarian.

Reviewer 2 ·

Basic reporting

Mikò and colleagues present a morphologically and taxonomically comprehensive manuscript. The paper uses a new fossil species as a kind of catalyst for assessing megaspilid and ceraphronoid synapomorphies and taxonomic assignments more broadly. The authors are clear experts in the morphology of the group and provide a nice description from a morphological point of view. I recommend the manuscript for publication, following that the authors respond to the following comments. Most importantly, the authors must improve the text relating to the age of Baltic amber and provide the geological setting in the description.

Line 36: "phylogenetic studies have failed to classify the superfamily with confidence" do you mean that the molecular studies do not support monophyly? If so, it would read more clearly with respect to phylogenetics vs taxonomy as "phylogenetic studies have failed to support the monophyly of the superfamily with confidence"

Line 38: Corresponding with above, "Irrespective of their phylogenetic position, recent ceraphronoids comprise a morphologically well-characterized group that can be readily separated from other hymenopterans based on the following traits." But, you (may) acknowledge that there may not be molecular support for the monophyly of Ceraphronoidea in the previous sentence. It would be helpful for overall clarity and consistency if you acknowledged the possibility of this similarity representing homoplasy (in cases where the molecular results indicate polyphyly) or perhaps a retention of an ancestral traits (if the group is recovered as paraphyletic).

Line 69: Very important: Molecular folks may be inclined to use your paper for future divergence estimations, so the age you report here should be accurate. I do not believe this age for Baltic amber is correct, I recommend investigating this age before reporting and absolutely recommend including a citation, for example with the following paper, which reports the age as ~44.1 Ma. Weitschat, W. & Wichard, W. 2010. Baltic amber. Penney, D. (Ed.): Biodiversity of fossils in amber from the major world deposits. Siri Scientific Press, Manchester, pp. 80-115. Also, worth mentioning that this fossil is from the Eocene somewhere in the paper and description or even title. Curiously, the age appears to be correct in supplemental figure 1.

Figure 1: It may be helpful for folks unfamiliar with the group to highlight sutures of the fused mesonotal sclerite and the individual components: pronotum, metanotum, mesopectus, etc. If difficult or you feel it would not contribute, feel free to ignore.

Figure 2: It would be nice to link boxes D and E in Fig 1 by showing where on the wing the triangular elements shown in the SEM image are found. This can be done with a bounding box in figure D, for example. Also just a recommendation.


Lines 165-251: The description would be improved with the following additional sections: etymology; locality and geographic range/age for the type material.

Line 272: if you feel it would contribute to your discussion: how plastic are flagellomere and pedicel lengths in other ceraphronoid taxa? If highly homoplastic elsewhere, this could help explain the differences as well.

Lines 324-325: These four traits do seem to be relatively homoplastic overall - if possible, recommend elaborating on why the flattened coxa, elongate spurs, and shortened metasoma in particular are not likely to be highly variable.

Supplemental figure 1: should have citations to give credit to original taxonomic papers as well as allow others to investigate the fossil history of the group in more detail.

Supplemental figure 1: the increments on the timeline are currently not labeled, and simply go from "1 million" at the far right (not present?) to 150. Please label at increments to make more legible.

Supplemental figure 1: what do the bracketed letters indicate? Apologies if this is in a figure caption not visible to reviewers.

Abstract: "Conostigmus talamasi Mikó and Trietsch, sp. nov. from Baltic amber represents an intermediate form between the type genus, Megaspilus, and one of the most species-rich megaspilid genera, Conostigmus, supporting the hypothesis that Megaspilus is nested within Conostigmus." There is another possibility for this similarity that does not reflect a nested relationship between these two genera: What if C. talamasi represents an early lineage of Conostigmus and so possesses a mosaic of Conostigmus and Megaspilus features for that reason?

Abstract: The abstract directly states that three taxa are removed from Ceraphronoidea: Dendrocerus dubitatus, Stigmaphronidae, and Radiophronidae. It appears that only one of these taxa are explicitly stated as being interpreted as incertae sedis in the main text (Radiophronidae). Dendrocerus dubitatus is not mentioned in the discussion text in this context, however Lygocerus dubitatus is - did I miss a synonymization or is this a typo of some kind?

Very minor suggestions:
Line 33: "Myriad other…" should perhaps be "A myriad of other…"

Line 45: Recommend avoiding "evolutionary advantage" as the fusion seen may be a requirement based on some other morphological feature or developmental requirement, rather than an advantage itself per se.

Line 57/58: Would read a little more clearly if "albeit" were replaced with "while"; "is present" should be "are present"

Line 69: "key specimens" could be "type material" to be more specific

Line 304: "detailed description as it was already outlined by McKellar and Engel" should be "detailed description as outlined by McKellar and Engel" or similar for clarity.

Line 340: "phenotype" refers to the overall composite of morphological features, recommend trait/character/feature instead.

Experimental design

Line 86: "In such poor conditions, only those who have trained their eyes by looking at thousands of ceraphronoid specimens are able to identify these taxa correctly." This seems a little excessive as a qualifier (would looking at hundreds of specimens be enough, for example) and not necessary in the context of the surrounding text.

Line 88: "In some ways, the examination of fossil specimens is similar to studying specimens of recent taxa with a low quality microscope. Morphological traits are usually obscured by artifacts or debris, making them difficult to properly observe (if they can be seen at all)." These sentences are true in some cases, but I'm not sure I agree with this overall. Amber fossils can preserve remarkable detail, and can even provide clean views of lightly sclerotized anatomy, for example by fixing inclusions with mouthparts exuded or preserving effectively "cleared" insects and exposing internal apodemes, etc. Not necessary to revise of course but putting in a conditional "in some cases" for poor preservation and obscure views would be more accurate in my view.

Line 105: "3D measurement tool Amira" should be "3D measurement tool in Amira" as Amira is the overall software package for segmenting, not just a measurement tool.

Line 119: was any darkening of the amber matrix noticed after the scan? This is typically the case with amber in the synchrotron and not reported, however it'd be helpful to report when this happens (only if it happened in this case, of course!).

Validity of the findings

no comment

---

## Round 0.2 · accepted · Accept

The authors have done a very good job addressing reviewer's comments and the paper is ready go on to the next stage of production.

#